# Optogenetic induction of appetitive and aversive taste memories in *Drosophila*

Meghan Jelen, Pierre-Yves Musso, Pierre Junca, Michael D Gordon*

Department of Zoology and Life Sciences Institute, University of British Columbia, Vancouver, Canada

**Abstract** Tastes typically evoke innate behavioral responses that can be broadly categorized as acceptance or rejection. However, research in *Drosophila melanogaster* indicates that taste responses also exhibit plasticity through experience-dependent changes in mushroom body circuits. In this study, we develop a novel taste learning paradigm using closed-loop optogenetics. We find that appetitive and aversive taste memories can be formed by pairing gustatory stimuli with opto-genetic activation of sensory neurons or dopaminergic neurons encoding reward or punishment. As with olfactory memories, distinct dopaminergic subpopulations drive the parallel formation of short- and long-term appetitive memories. Long-term memories are protein synthesis-dependent and have energetic requirements that are satisfied by a variety of caloric food sources or by direct stimulation of MB-MP1 dopaminergic neurons. Our paradigm affords new opportunities to probe plasticity mechanisms within the taste system and understand the extent to which taste responses depend on experience.

## Editor's evaluation

Through a new operant learning assay and fly genetics, this important work convincingly shows that taste memory formation requires the same circuit substrates and mechanisms as olfactory memory formation. While the exact mechanisms remain to be elucidated, the convincing data and approach represent a valuable foundation for the study of molecular and circuit mechanism underpinning taste memory formation and the role of brain energy therein. This study will be of particular interest to the large community of scientists studying the mechanisms and circuits of memory formation in the fly and possibly beyond.

**\*For correspondence:**
michael.gordon@ubc.ca

**Competing interest:** The authors declare that no competing interests exist.

## Introduction

Food selection is influenced by a complex set of factors including external sensory input, interocep-tive circuits signaling internal state, and plasticity driven by past feeding experiences. The gustatory system plays a critical role in evaluating the nutritional qualities of foods, and is generally thought to evoke innate appetitive or aversive behavioral responses. However, the degree to which taste processing can be modified by learning is unclear.

In flies, taste detection is mediated by gustatory receptor neurons (GRNs) located on the proboscis, pharynx, legs, wing margins, and ovipositor (*Stocker, 1994*). GRNs express a range of chemosensory receptors for detecting sugars, bitters, salts, and other contact chemical cues (*Chen and Dahanukar, 2020*). GRNs project to the subesophageal zone (SEZ) of the fly brain, where taste information is segregated based on modality, valence, and organ of detection (*Marella et al., 2006*; *Thorne et al., 2004*; *Wang et al., 2004*).

Although the valence of a specific taste is generally set, the intensity of the response can vary substantially according to internal state. Starvation increases a fly's sensitivity to sweet tastes and

blunts bitter responses through direct modulation of GRN activity (*Inagaki et al., 2014*; *Inagaki et al., 2012*; *LeDue et al., 2016*; *Marella et al., 2012*). Moreover, flies lacking essential nutrients such as amino acids and salts exhibit increased nutrient-specific preference toward foods containing those substances (*Corrales-Carvajal et al., 2016*; *Jaeger et al., 2018*; *Steck et al., 2018*).

In addition to internal state-dependent changes in nutrient drive, fly taste responses can be altered by experience. Most notably, short-term taste-specific suppression of appetitive responses can be achieved through pairing the appetitive taste with either bitter stimulation or noxious heat (*Keene and Masek, 2012*; *Kirkhart and Scott, 2015*; *Masek et al., 2015*; *Tauber et al., 2017*). This plasticity requires an integrative memory association area called the mushroom body (MB), which is known to represent different sensory modalities, including olfaction and taste (*Cohn et al., 2015*; *Davis, 2005*; *Keene and Masek, 2012*; *Keene and Waddell, 2007*; *Kirkhart and Scott, 2015*; *Masek et al., 2015*; *Schwaerzel et al., 2003*). Thus, while taste responses are executed by innate circuits, they also exhibit experience-dependent changes driven by the adaptable networks of the MBs (*Colomb et al., 2009*; *Kirkhart and Scott, 2015*; *Krashes et al., 2009*).

The MBs are composed of approximately ~4000 intrinsic Kenyon cells (KCs), whose dendrites receive inputs from different sensory systems (*Kirkhart and Scott, 2015*; *Schwaerzel et al., 2003*; *Tanaka et al., 2008*; *Vogt et al., 2014*). KCs form *en passant* synapses with mushroom body output neurons (MBONs), and MBONs send projections to neuropils outside of the MBs to modulate behavior (*Crittenden et al., 1998*; *Tanaka et al., 2008*). Each MBON receives KC input in a specific region of the MB called a 'compartment', and activation of an MBON typically evokes either a positive (approach) or negative (avoidance) valence (*Perisse et al., 2013*). Integration across many MBON responses is thought to produce the final behavioral valence and intensity (*Aso et al., 2014*).

Much of what we know about the mechanisms of associative memory formation in the MB comes from studies pairing olfactory stimuli with either sugar (reward) or electric shock (punishment). In these paradigms, an odor serves as the conditioned stimulus (CS) and produces sparse activation of a unique combination of KCs (*Beck et al., 2000*; *Tempel et al., 1983*; *Tully, 1984*; *Tully and Quinn, 1985*). Meanwhile, sugar or electric shock serves as the unconditioned stimulus (US) by evoking activity in distinct populations of dopaminergic neurons (DANs) – protocerebral anterior medial (PAM) DANs are activated by sugar, while protocerebral posterior lateral 1 (PPL1) DANs are activated by shock (*Burke and Waddell, 2011*; *Gervasi et al., 2010*; *Mao and Davis, 2009*; *Tomchik and Davis, 2009*). DANs target specific MB compartments, where dopamine functions to depress the synaptic connections between active KCs and the compartment's MBONs (*Aso et al., 2014*; *Cohn et al., 2015*; *Hige et al., 2015*; *Perisse et al., 2013*). Strikingly, rewarding PAM DANs generally target compartments with MBONs carrying negative valence, while PPL1 DANs target compartments with MBONs carrying positive valence. Thus, the resulting change in synaptic weights following concurrent activation of KCs with either PAM or PPL1 skews behavior toward either approach or avoidance (*Aso et al., 2010*; *Perisse et al., 2013*).

Consistent with this model, direct activation of DANs in the absence of any rewarding or punishing stimulus can function as a US in some fly associative learning paradigms (*Aso et al., 2012*; *Claridge-Chang et al., 2009*; *Colomb et al., 2009*; *Liu et al., 2012*). Optogenetic or thermogenetic activation of PAM DANs following, or in coincidence with, an odor results in the formation of an appetitive memory. Meanwhile, activation of punishing PPL1 DANs leads to the formation of an aversive memory (*Cohn et al., 2015*; *Yamagata et al., 2015*). A similar phenomenon has also been demonstrated in mice, where phasic optogenetic activation of specific dopaminergic subsets can lead to the formation of conditioned behaviors, even in the absence of a physical reward (*Saunders et al., 2018*).

DAN populations are also segregated by the type of memory formed, as appetitive short-term memories (STM) and long-term memories (LTM) are formed by independent PAM subpopulations (*Burke and Waddell, 2011*; *Colomb et al., 2009*; *Musso et al., 2015*; *Yamagata et al., 2015*). Moreover, whereas STM may be formed with a sweet tasting reward on its own, the formation of LTM requires a sweet and nutritious US (*Burke and Waddell, 2011*; *Musso et al., 2015*). Caloric sugars are thought to gate memory consolidation by promoting sustained rhythmic activity of MB-MP1 DANs (*Musso et al., 2015*; *Plaçais et al., 2017*; *Plaçais et al., 2012*). Interestingly, this signaling may occur up to 5 hr post ingestion, suggesting that there is a critical time window for the formation of LTM (*Musso et al., 2015*; *Pavlowsky et al., 2018*).

Although flies are known to exhibit aversive short-term taste memories, the full extent to which taste behaviors are modifiable by learning is unknown. Can taste responses be enhanced by appetitive conditioning? Can flies form LTM about taste? These are difficult questions to answer using traditional methods for several reasons. First, appetitive association paradigms generally rely on food as the US, which interferes with the representation of a taste CS and can also modify future taste behaviors through changes in satiety state. Second, taste is an active sense, and animals typically have behavioral control over exposure to the stimulus. Thus, repeated temporal pairing of a taste CS with a US is difficult to achieve in flies without immobilization, making LTM difficult to test. Moreover, the self-control over taste exposure under more natural conditions could support operant learning with neural and molecular mechanisms that are distinct from classic olfactory conditioning (*Brembs, 2009*).

To probe the potential of taste learning, we developed an optogenetic learning paradigm that couples a taste (the CS) with optogenetic GRN or DAN stimulation (the US). Using this novel paradigm, we show that flies can form both appetitive and aversive short- and long-term taste memories. As in olfaction, appetitive taste memories are driven by discrete PAM populations, and activation of a single PAM subpopulation is sufficient to induce appetitive LTM. The formation of appetitive LTM requires de novo protein synthesis and is contingent on caloric intake. Moreover, sugar, certain amino acids, and lactic acid can provide the energy required to support LTM formation, and this requirement is also satisfied by thermogenetic activation of MB-MP1 neurons.

## Results

### Pairing GRN activation with a food source leads to taste memory formation

We previously developed a system called the sip-triggered optogenetic behavioral enclosure (STROBE), in which individual flies are placed in an arena with free access to two odorless food sources (*Musso et al., 2019*). Interactions (mostly sips) with one food source triggers nearly instantaneous activation of a red LED, which can be used for optogenetic stimulation of neurons expressing CsChrimson. We reasoned that sipping on a tastant (the CS+) that triggers activation of neurons providing either positive or negative reinforcement may produce a change in the number of interactions a fly initiates upon subsequent exposure to the same CS+ (*Figure 1A*).

We began by testing the efficacy of the STROBE in inducing aversive and appetitive memories through optogenetic activation of bitter and sweet GRNs, respectively. Bitter GRN stimulation is known to activate PPL1 DANs, while sweet GRNs activate PAMs (*Keene and Masek, 2012*; *Kirkhart and Scott, 2015*; *Liu et al., 2012*; *Masek et al., 2015*). Moreover, bitter or sweet GRN activation with *Gr66a-* or *Gr43a-Gal4* is sufficient for STM induction in taste and olfactory associative learning paradigms (*Keene and Masek, 2012*; *Yamagata et al., 2015*). Therefore, we tested whether pairing GRN activation with feeding on a single taste modality could create an associative taste memory that altered subsequent behavior to the taste.

In the aversive taste memory paradigm, interactions with 25 mM sucrose (CS+) during training triggered LED activation of Gr66a bitter neurons expressing CsChrimson (*Figure 1—figure supplement 1A*). This led to CS+ avoidance relative to plain agar (CS-) during training (*Figure 1B*). During testing, we disabled the STROBE lights and measured preference toward 25 mM sucrose (CS+) relative to agar (CS-) to see if flies have formed aversive taste memories. Indeed, 10 min after training, flies that experienced bitter GRN activation during training showed a lower sugar preference than control flies lacking the obligate CsChrimson cofactor all-*trans*-retinal or not expressing CsChrimson. Like most of the experiments that will follow, there was high variance in the behavior of individual flies during both training and testing, undoubtedly reflecting a combination of individual variation in internal state, past experiences, response to training, as well as stochastic effects during the measurement time period. Examining the preference indices over time revealed that the difference in preference emerged after about 30 min of testing, which could either reflect a progressive divergence in behavior between the groups or, more likely, increased reliability of the preference measurement as sips accumulate over time (*Figure 1C*). A similar aversive memory was also produced by the activation of PPK23^glut 'high salt' GRNs, which carry a negative valence in salt-satiated flies (*Figure 1—figure supplement 1A, B*). Importantly, these effects are not due to heightened satiety in trained flies, because training in this

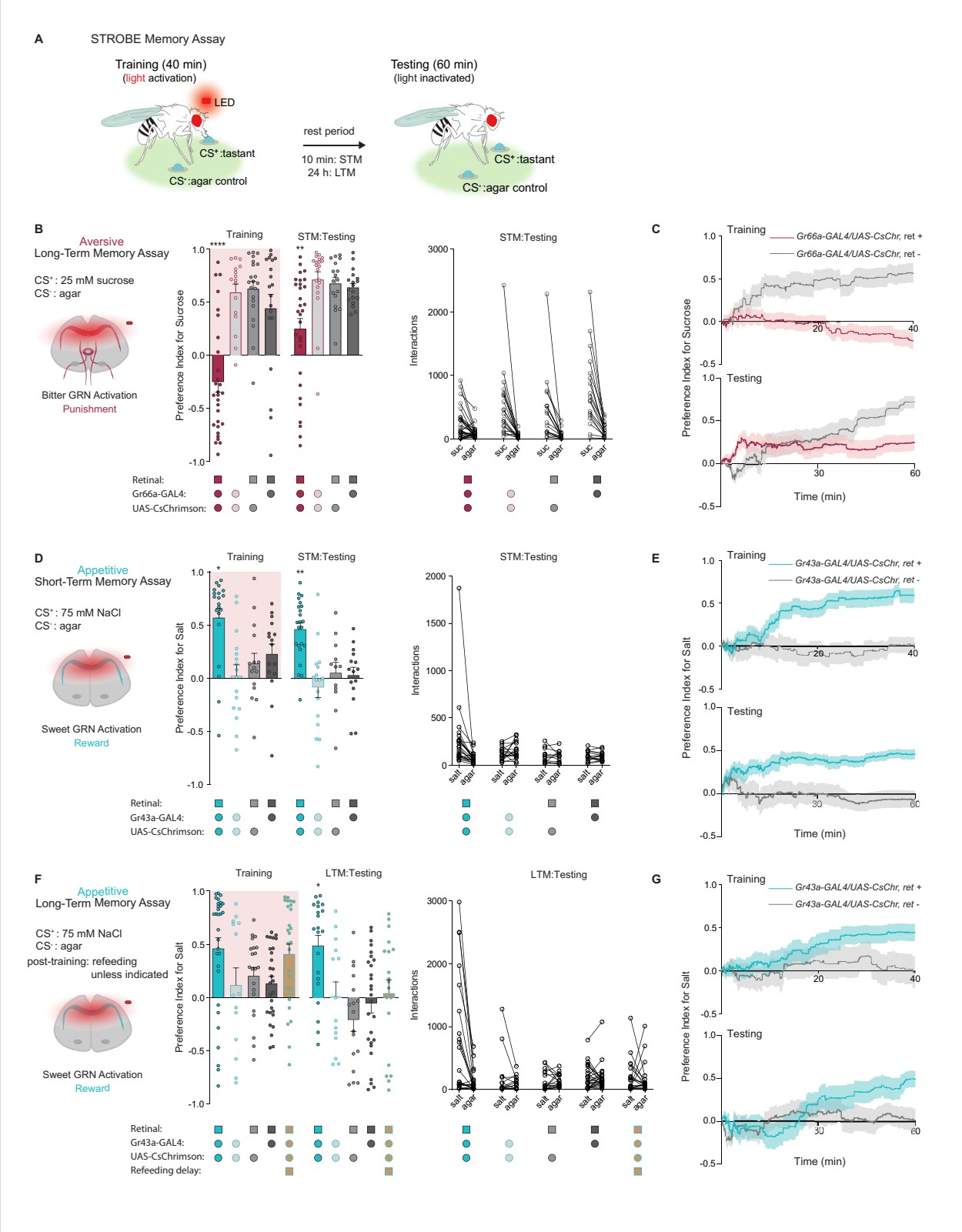

**Figure 1.** Gustatory receptor neurons (GRNs) produce punishment and reward signals capable of facilitating taste memory formation. (**A**) Diagram outlining sip-triggered optogenetic behavioral enclosure (STROBE) memory paradigm. Training: 24 hr starved flies freely interact with a LED-activating tastant (CS⁺) and a non-LED-activating tastant (CS⁻) for 40 min. LED activation stimulates *CsChrimson*-expressing neurons. Testing: associative memory is measured by assessing a fly's preference for the CS⁺ tastant compared to the CS⁻ for 1 hr. In the short-term memory (STM) assay testing occurs 10 min

*Figure 1 continued on next page*

*Figure 1 continued*

after training. In the long-term memory (LTM) assay testing occurs 24 hr after training. (**B**) Aversive STM measured after pairing 25 mM sucrose (CS$^+$) with bitter neuron optogenetic activation. Preference indices (left) and tastant interactions (right) for *Gr66a>CsChrimson* flies compared to controls during training and testing. The interaction numbers for individual flies are connected by lines. (**C**) Cumulative average preference indices over the course of training and testing in (**B**), (n=16–30). (**D**) Appetitive STM measured after pairing 75 mM NaCl (CS$^+$) with sweet neuron optogenetic activation. Preference indices (left) and interactions (right) for *Gr43a>CsChrimson* flies compared to controls in the short-term memory assay. (**E**) Preference index of flies in (**D**) over time during training and testing (n=12–23). (**F**) Appetitive LTM measured after pairing of 75 mM NaCl (CS$^+$) with sweet neuron optogenetic activation. Preference indices (left) and interactions (right) for *Gr43a>CsChrimson* flies compared to controls in the LTM assay. (**G**) Average preference index as a function of time for the training and testing in the LTM assay (n=14–30). All flies were starved for 24 hr prior to training. Preference index is mean ± SEM, Kruskal-Wallis with Dunn's multiple comparison test: **p < 0.01, ****p < 0.0001.

The online version of this article includes the following source data and figure supplement(s) for figure 1:

**Source data 1.** Raw data for *Figure 1*.

**Figure supplement 1.** Gustatory receptor neuron (GRN) activation produces reward and punishment signals in taste memory formation.

**Figure supplement 1—source data 1.** Raw data for *Figure 1—figure supplement 1*.

paradigm is associated with fewer food interactions than controls (*Figure 1C* and *Figure 1—figure supplement 1B*).

For appetitive training, we chose 75 mM NaCl as the CS$^+$, since flies show neither strong attraction nor aversion to this concentration of salt (*Zhang et al., 2013*). Interactions with the CS$^+$ in this paradigm triggered optogenetic activation of sweet neurons, either with *Gr43a-Gal4*, which labels a subset of leg and pharyngeal sweet neurons in addition to fructose-sensitive neurons in the protocerebrum, or *Gr64f-Gal4*, which labels most peripheral sweet GRNs (*Figure 1—figure supplement 1A*). In both cases, sweet GRN activation produced an increased preference for the salt CS$^+$ during training and testing 10 min later (*Figure 1D* and *Figure 1—figure supplement 1C*). The increased preference is evident early during testing and maintained throughout the testing phase (*Figure 1*). Like the aversive memory paradigm, the effects of appetitive conditioning cannot easily be explained through changes in internal state, since trained flies interacted more with the food during training and therefore should have a lower salt drive during testing. Interestingly, refeeding flies with standard medium directly after training in the appetitive paradigm led to a long-term preference for the CS$^+$, revealed by testing 24 hr later (*Figure 1F and G* and *Figure 1—figure supplement 1D*). This stands in contrast to the aversive paradigm, where reduced preference for sugar following bitter GRN activation was absent 24 hr later (*Figure 1—figure supplement 1A, E*).

## DAN activation is sufficient for the induction of short- and long-term taste memories

We next asked whether direct activation of DANs during feeding could drive the formation of taste memories. Aversive short-term taste memory depends on multiple PPL1 DANs, including PPL1-α'2 α2 and PPL1-α3 (*Masek et al., 2015*), while appetitive short-term taste memories have not been previously reported. We first tested whether activating PPL1 DANs coincident with tastant interactions would lead to STM formation in the STROBE. Stimulation of PPL1 neurons reduced sucrose preference during training, and a reduced preference was also observed during STM testing 10 min later (*Figure 2A*). This decreased preference was sustained throughout the entire period of testing (*Figure 2B*). Interestingly, unlike activation of bitter sensory neurons, PPL1 activation also produced a long-term aversive memory that was expressed 24 hr after training and remained stable through the duration of testing (*Figure 2C and D*).

To test the effect of appetitive DAN activation, we used flies expressing CsChrimson in PAM neurons under control of the broad PAM driver *R58E02-Gal4*. Intriguingly, although optogenetic activation of PAM neurons signals reward to the MB, it did not affect immediate preference toward light-paired 75 mM NaCl (CS$^+$) during training. Nonetheless, this pairing resulted in appetitive memory expression during testing 10 min and 24 hr after training (*Figure 2E and G*). These taste memories were stable throughout the entire duration of testing (*Figure 2F and H*). Thus, optogenetic activation of PAM neurons in the STROBE was able to write both short- and long-term appetitive taste memories in the absence of acute effects on feeding.

Given that the taste memories we observe are created in a novel and uncharacterized paradigm, we did additional experiments with PAM activation to establish that the memories were specific to

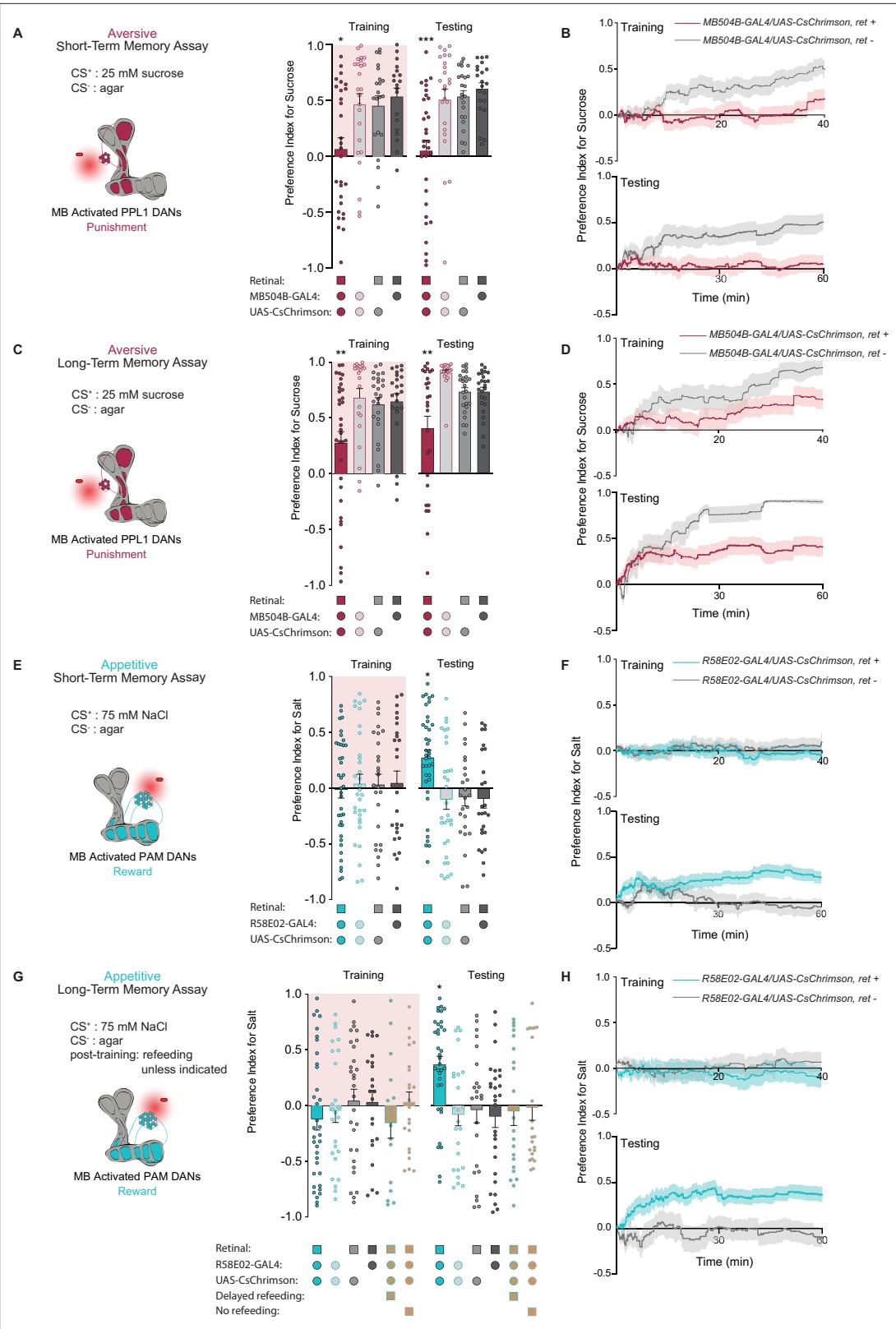

**Figure 2.** PPL1 and protocerebral anterior medial (PAM) neural activation is sufficient for the induction of short- and long-term taste memories. (**A**) Aversive short-term memories (STM) measured following PPL1 neuron optogenetic activation paired with 25 mM sucrose (CS⁺) vs agar (CS⁻). Flies lacking retinal or one genetic element for expression of CsChrimson serve as controls (n=19–31). (**B**) Preference indices over time for the experiment shown in (**A**). (**C**) Aversive long-term memories (LTM) measured following PPL1 optogenetic activation paired with 25 mM sucrose (CS⁺) vs agar (CS⁻)

*Figure 2 continued on next page*

*Figure 2 continued*

(n=20–33). (**D**) Preference indices over time for the experiment shown in (**C**). (**E**) Appetitive STM measured following PAM neuron optogenetic activation paired with 75 mM NaCl (CS⁺) vs agar (CS⁻) (n=25–38). (**F**) Preference indices over time for the experiment shown in (**E**). (**G, H**) Appetitive LTM measured following PAM neuron optogenetic activation paired with 75 mM NaCl (CS⁺) vs agar (CS⁻) (n=17–35). Flies were refed with standard food for 1 hr directly after training unless otherwise indicated as delayed refeeding (8 hr after training) or no refeeding. (**H**) Preference indices over time for the experiment shown in (**G**). All flies were food deprived for 24 hr prior to the start of experimentation. Preference indices are mean ± SEM, Kruskal-Wallis with Dunn's multiple comparison test: *p<0.05, **p<0.01, ***p<0.001.

The online version of this article includes the following source data and figure supplement(s) for figure 2:

**Source data 1.** Raw data for *Figure 2*.

**Figure supplement 1.** Taste memories are specific to the CS⁺.

**Figure supplement 1—source data 1.** Raw data for *Figure 2—figure supplement 1*.

---

the CS⁺ tastant. First, flies trained with NaCl as the CS⁺ and agar as the CS⁻ showed no preference between two identical agar options during testing, ruling out the possibility that the increased CS⁺ preference observed in prior experiments was driven by a spatial memory or other non-CS⁺ local cues such as deposited pheromones (*Figure 2—figure supplement 1A*). This remains true when using sweet sensory neuron activation as the US, which, unlike PAM stimulation, drives elevated preference for the salt option during training (*Figure 2—figure supplement 1B*). Next, we found that a second tastant, monopotassium glutamate (MPG), could replace NaCl as the CS⁺. MPG is approximately equally appetitive to NaCl (*Figure 2—figure supplement 1C*), and pairing of MPG with PAM activation resulted in a robust appetitive memory to MPG (*Figure 2—figure supplement 1D*). Moreover, training with NaCl as the CS⁺ and MPG as the CS⁻ produced an appetitive memory for NaCl (*Figure 2—figure supplement 1E*). Finally, flies trained with NaCl as the CS⁺ and agar as the CS⁻ did not show elevated preference for MPG introduced as a novel tastant during testing (*Figure 2—figure supplement 1F*). All these observations support the conclusion that memories formed during STROBE training are taste memories specific to the trained CS⁺.

We also sought to establish the energy requirements for appetitive LTM formed through PAM activation. Based on the critical role of energy in long-term olfactory memory formation (*Musso et al., 2015*; *Plaçais et al., 2017*; *Plaçais et al., 2012*), we designed our LTM paradigm to include a brief 1 hr exposure to food after training. To confirm the necessity of this feeding, we tested flies that were not fed after training or were fed 7 hr post training, after the memory consolidation time period defined in olfactory memory (*Figure 2G*). Neither of these groups expressed taste memories during testing. Thus, the contingencies governing the formation and expression of taste memories in *Drosophila* appear similar to those previously discovered for olfaction.

## The MBs are required for short- and long-term taste memory formation

The intrinsic neurons of the MB are required for aversive taste memory formation (*Masek et al., 2015*). To demonstrate that the MBs are also required for appetitive taste memory formation, we silenced KCs throughout both our STM and LTM assays using tetanus toxin expressed under control of the pan-KC driver *R13F02-LexA*. KC silencing eliminated both short-term and long-term appetitive memories formed by activation of Gr43a sensory neurons (*Figure 3A and B*) or PAM neurons labeled by the driver *R58E02-Gal4* (*Figure 3C and D*). These findings indicate that MB intrinsic neurons play a pivotal role in the formation of appetitive taste memories.

Olfactory LTM requires de novo protein synthesis during memory consolidation (*Colomb et al., 2009*). To test whether the same is true for taste memories, we fed flies the protein synthesis inhibitor cycloheximide (CXM). As expected, flies fed CXM prior to training were unable to form long-term taste memories, in contrast to vehicle controls (*Figure 3E*). These results confirm that the taste memories being formed are protein synthesis dependent, consistent with the classic characteristics of LTM (*Figure 3F*).

## Distinct PAM subpopulations induce appetitive short- and long-term taste memories

Distinct subpopulations of PAM neurons – those targeting β′2, γ4, and γ5 compartments labeled by *R48B04-Gal4* and those targeting α1, β′1, β2, and γ5 compartments labeled by *R15A04-Gal4* – mediate

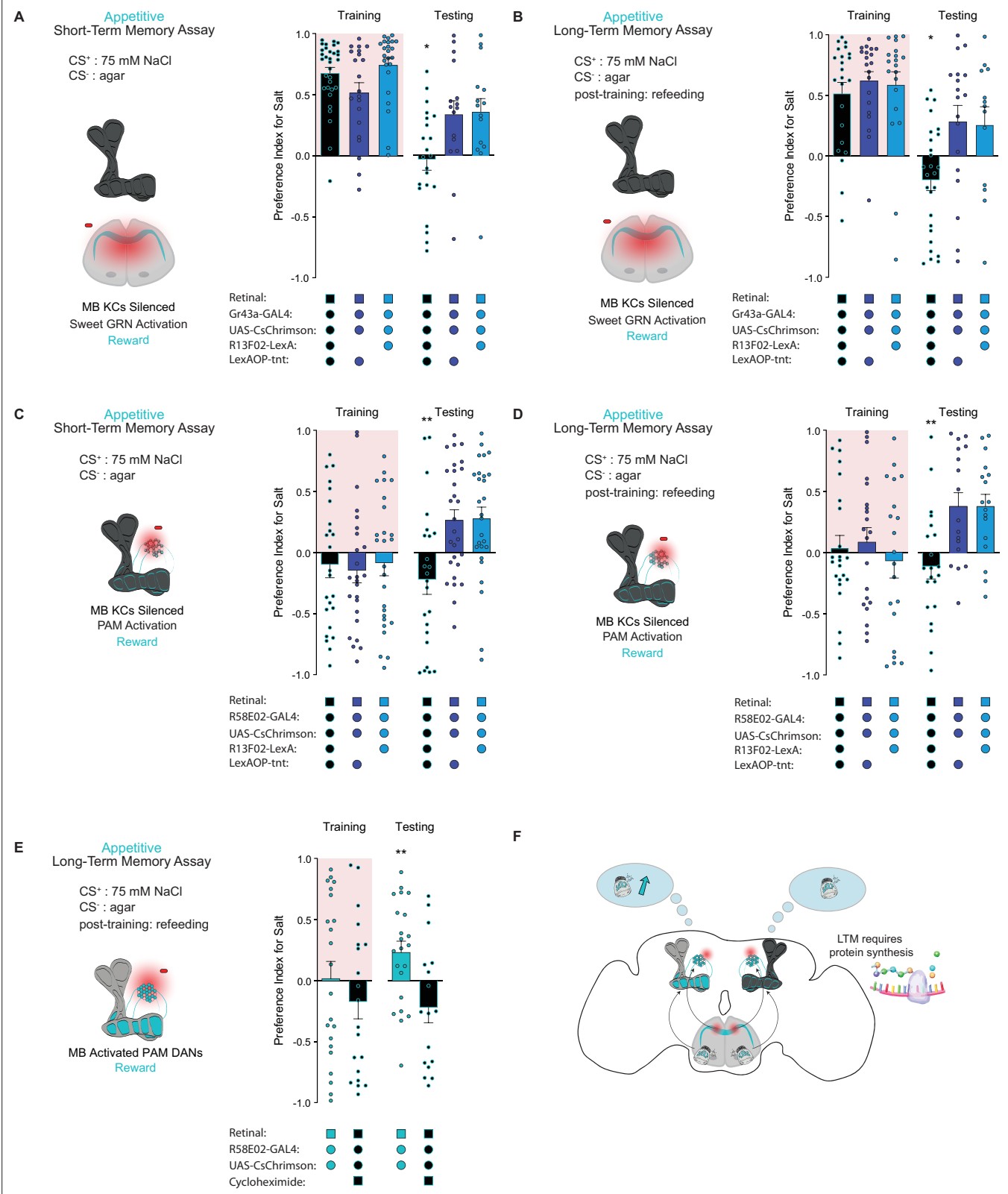

**Figure 3.** The mushroom body (MB) is required for the formation of short- and long-term taste memories. (**A, B**) Appetitive short-term memories (STM) (**A**) and long-term memories (LTM) (**B**) measured following sweet taste neuron optogenetic activation paired with 75 mM NaCl (CS⁺) vs agar (CS⁻) with Kenyon cells (KCs) silenced by expression of tetanus toxin (n=16–34 for STM and n=13–27 for LTM). Controls are missing one genetic element for KC silencing and therefore exhibit memory. (**C, D**) Appetitive STM (**C**) and LTM (**D**) measured following protocerebral anterior medial (PAM) neuron

*Figure 3 continued on next page*

*Figure 3 continued*

optogenetic activation paired with 75 mM NaCl (CS⁺) vs agar (CS⁻) with KCs silenced by expression of tetanus toxin (n=24–28 for STM and n=17–23 for LTM). Controls are missing one genetic element for KC silencing and therefore exhibit memory. Memory assays when the MB is silenced, compared to controls. (**E**) Appetitive LTM measured following PAM neuron optogenetic activation with 75 mM NaCl (CS⁺) vs agar (CS⁻). Flies were either fed retinal or retinal plus cycloheximide (n=17–22). (**F**) Model of appetitive taste memory formation via gustatory receptor neuron (GRN)/PAM activation. All flies were starved for 24 hr prior to training. Preference indices are mean ± SEM, Kruskal-Wallis with Dunn's multiple comparison test (**A–D**) or Mann-Whitney test (**E**): *p < 0.05, **p < 0.01.

The online version of this article includes the following source data for figure 3:

**Source data 1.** Raw data for *Figure 3*.

the formation of appetitive short- and long-term olfactory memories, respectively (*Yamagata et al., 2015*). Moreover, it has been hypothesized that two differential reinforcing effects of sugar reward – sweet taste and nutrition – are encoded by these segregated STM and LTM neural populations (*Yamagata et al., 2015*). We tested both populations in our appetitive STROBE memory assays to determine if the activation of these separate PAM clusters would support the formation of parallel short- and long-term taste memories. Indeed, activation of the β′2, γ4, and γ5 regions drove appetitive short-term but not long-term taste memories, as shown by the higher salt preference of flies expressing active CsChrimson during STM testing but not LTM testing (*Figure 4A and B*). Conversely, activation of the α1, β′1, β2, and γ5 compartments produced LTM but not STM (*Figure 4C and D*). These results indicate that, much like appetitive olfactory memory, short- and long-term taste memories are formed by distinct PAM subpopulations.

Next, we wondered whether activation of a single PAM cell subtype, PAM-α1, would be sufficient to induce taste memories. PAM-α1 neurons project to an MB compartment innervated by MBON-α1, which in turn feeds back onto PAM-α1 to form a recurrent reward loop necessary for the formation of appetitive olfactory LTM (*Aso and Rubin, 2016*; *Ichinose et al., 2015*). Consistent with its role in olfactory memory, activation of this PAM cell type in the STROBE with drivers *MB043B-Gal4* or *MB299B-Gal4* was sufficient to drive appetitive long-term, but not short-term, taste memory formation (*Figure 4E and F* and *Figure 4—figure supplement 1A, B*).

Interestingly, activation of the PAM-β2β′2a subset labeled by *MB301B-Gal4* produced a higher preference for the salt CS during training, yet no sustained changes in taste preference during STM or LTM testing were observed (*Figure 4G and H*). This demonstrates that the reward signaling associated with PAM cell activation occurs on multiple timescales to produce acute, short-, or long-term changes in behavior, consistent with past results demonstrating the context-dependent effects of DAN activation (*Rohrsen et al., 2021*) Notably, the trend toward lower salt preference during testing in this experiment may reflect a reduced salt drive due to increased salt consumption during training.

## Caloric food sources are required for the formation of associative long-term taste memories

Because refeeding with standard fly medium shortly after training is permissive for the consolidation of appetitive long-term taste memories, we next asked what types of nutrients support memory formation. As expected, refeeding with L-glucose, a non-caloric sugar, did not lead to formation of associative long-term taste memories (*Figure 5A and B*). However, along with sucrose, refeeding with lactic acid, yeast extract, and L-alanine promoted LTM, while L-aspartic acid did not. These results indicate that, in addition to sucrose, other caloric nutrients can provide sufficient energy for long-term taste memory formation. Moreover, 7 hr delayed refeeding of each nutrient failed to support memory formation (*Figure 5B*). Thus, similar to olfactory LTM, the formation of appetitive taste LTM is dependent on an energy source being readily available during the memory consolidation window (*Fujita and Tanimura, 2011*; *Musso et al., 2015*).

Our findings concerning the formation and expression of appetitive taste LTM bear striking similarities to those of olfactory LTM in terms of MB circuitry, dependence on protein synthesis, and energetic requirements. This led us to wonder if MB-MP1 neurons, which signal onto the MB and promote energy flux in MB neurons during LTM, perform a similar function in taste memory (*Musso et al., 2015*; *Plaçais et al., 2017*; *Plaçais et al., 2012*). To test this hypothesis, we activated MB-MP1 neurons directly after training using *UAS-TRPA1* and delayed refeeding to outside the memory consolidation window. Compared to genetic controls, flies in which MB-MP1 neurons were activated post

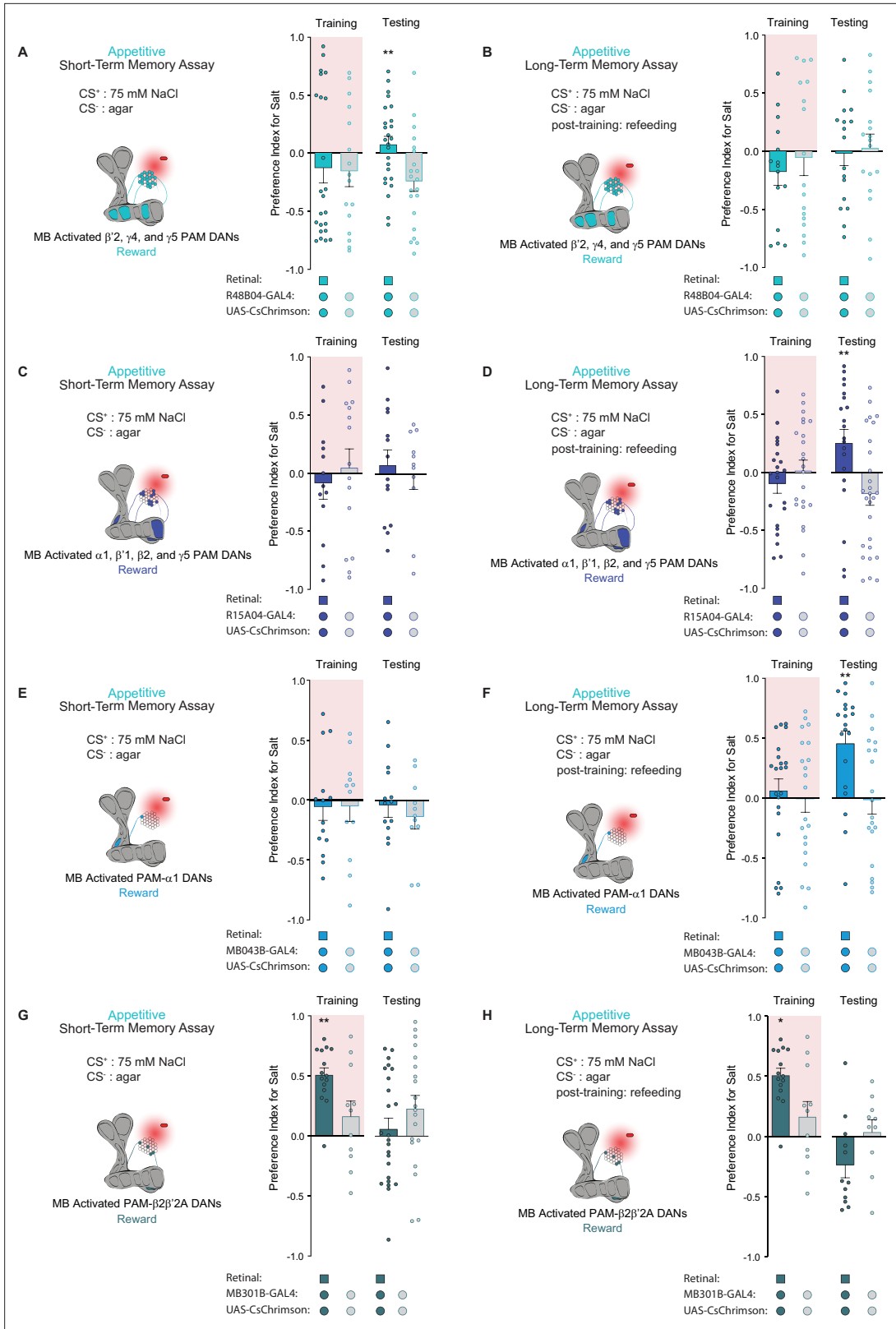

**Figure 4.** Distinct protocerebral anterior medial (PAM) subpopulations induce appetitive short- and long-term taste memories. (**A, B**) Appetitive short-term memories (STM) (**A**) and long-term memories (LTM) (**B**) measured following β'2, γ4, and γ5 PAM neuron optogenetic activation paired with 75 mM NaCl (CS+) vs agar (CS−) (n=21–28 for STM and n=15–17 for LTM). (**C, D**) Appetitive STM (**C**) and LTM (**D**) measured following α1, β'1, β2, and γ5 PAM neuron optogenetic activation paired with 75 mM NaCl (CS+) vs agar (CS−) (n=11–15 for STM and n=20–27 for LTM). (**E, F**) Appetitive STM (**E**) and LTM

*Figure 4 continued on next page*

*Figure 4 continued*

(**F**) measured following PAM-α1 neuron optogenetic activation paired with 75 mM NaCl (CS⁺) vs agar (CS⁻) (n=11–14 for STM and n=19–22 for LTM). (**G, H**) Appetitive STM (**G**) and LTM (**H**) measured following PAM-β2β'2a neuron optogenetic activation paired with 75 mM NaCl (CS⁺) vs agar (CS⁻) (n=20–27 for STM and n=10–15 for LTM). All flies were starved for 24 hr prior to training. Preference indices are mean ± SEM, Mann-Whitney test: **p < 0.01.

The online version of this article includes the following source data and figure supplement(s) for figure 4:

**Source data 1.** Raw data for *Figure 4*.

**Figure supplement 1.** Activation of discrete protocerebral anterior medial (PAM) subpopulations induces distinct types of taste memories.

**Figure supplement 1—source data 1.** Raw data for *Figure 4—figure supplement 1*.

training showed significantly elevated memory scores during testing (*Figure 6A and B*). This confirms that MB-MP1 activation is sufficient to drive memory consolidation during long-term appetitive taste memory formation (*Figure 6C*).

## Discussion

Gustation plays a vital role in determining the suitability of foods for ingestion. Yet, little is known about how experience influences higher-order taste representations and contributes to the continuous refinement of food selection. In fact, a memory system for the recollection of appetitive taste memories has not been described in flies. In this study, we use the STROBE to establish a novel learning paradigm and further investigate the formation and expression of taste memories. We demonstrate that flies can form short- and long-term appetitive and aversive taste memories toward two key nutrients – salt and sugar. Much like olfactory memory, associative taste memory formation occurs within the MB and follows many of the same circuit and energetic principles.

It is perhaps not surprising that olfactory and taste memories share common principles; however, important distinctions exist between olfactory and taste learning paradigms that justify the possibility that this may not have been the case. Most notably, using taste as a CS in a free feeding situation where reinforcement is temporally coupled to food contact creates the potential for operant, rather than classical, conditioning. These two types of learning can employ distinct neural circuits in rodents (*Ostlund and Balleine, 2007*) and are separable by their synaptic properties and molecular mechanisms in invertebrates (*Brembs, 2009*; *Brembs and Plendl, 2008*; *Hawkins and Byrne, 2015*). Nevertheless, when flies are tethered in a flight arena and punishment is predicted by a mixture of classical (color) and operant (self-motion) cues, the classical conditioning system overrides the operant conditioning system (*Brembs and Plendl, 2008*). In the STROBE, our data suggests that the fly learns the tastant as a classical cue, despite the operant component of the reinforcement contingencies. Thus, the conservation between olfactory and taste learning mechanisms is consistent with past studies.

Although aversive taste memories have been established, prior evidence for appetitive taste memories has been sparse. Rats' hedonic response to bitter compounds can be made more positive through pairing with sugar, and human studies suggest that children's taste palates are malleable based on positive experiences with bitter vegetables (*Breslin et al., 1990*; *Figueroa et al., 2020*; *Forestell and LoLordo, 2000*; *Wadhera et al., 2015*). Therefore, despite the difficulties of measuring taste memories in the lab, appetitive taste plasticity is very likely an ethologically important process.

We observed enhanced salt feeding following pairing of salt taste with sweet sensory neuron stimulation. This may be surprising, given that NaCl on its own activates sweet GRNs (*Jaeger et al., 2018*; *Marella et al., 2006*). However, 75 mM NaCl moderately activates only about one third of sweet GRNs (*Dweck et al., 2022*), and thus appetitive memory formation may be driven by strong activation of the broader sweet neuron population. Nevertheless, using direct stimulation of DANs as the US afforded us the ability to reduce this complication and also interrogate the roles of specific DAN populations. Taking a hypothesis-driven approach, we confirmed that PAM neural subpopulations reinforce taste percepts much like olfactory inputs, and that STM and LTM are processed by distinct subpopulations. For example, activating β'2, γ4, and γ5 compartments with *R48B04-Gal4* produces STM in both olfactory and taste paradigms, while activation of α1, β'1, β2, and γ5 with *R15A04-Gal4* produces LTM in both. These results confirm that appetitive short- and long-term taste memories are processed in parallel in the MB (*Trannoy et al., 2011*; *Yamagata et al., 2015*). Given that tastes, like odors, activate the KC calyces (*Kirkhart and Scott, 2015*), we speculate that optogenetic stimulation

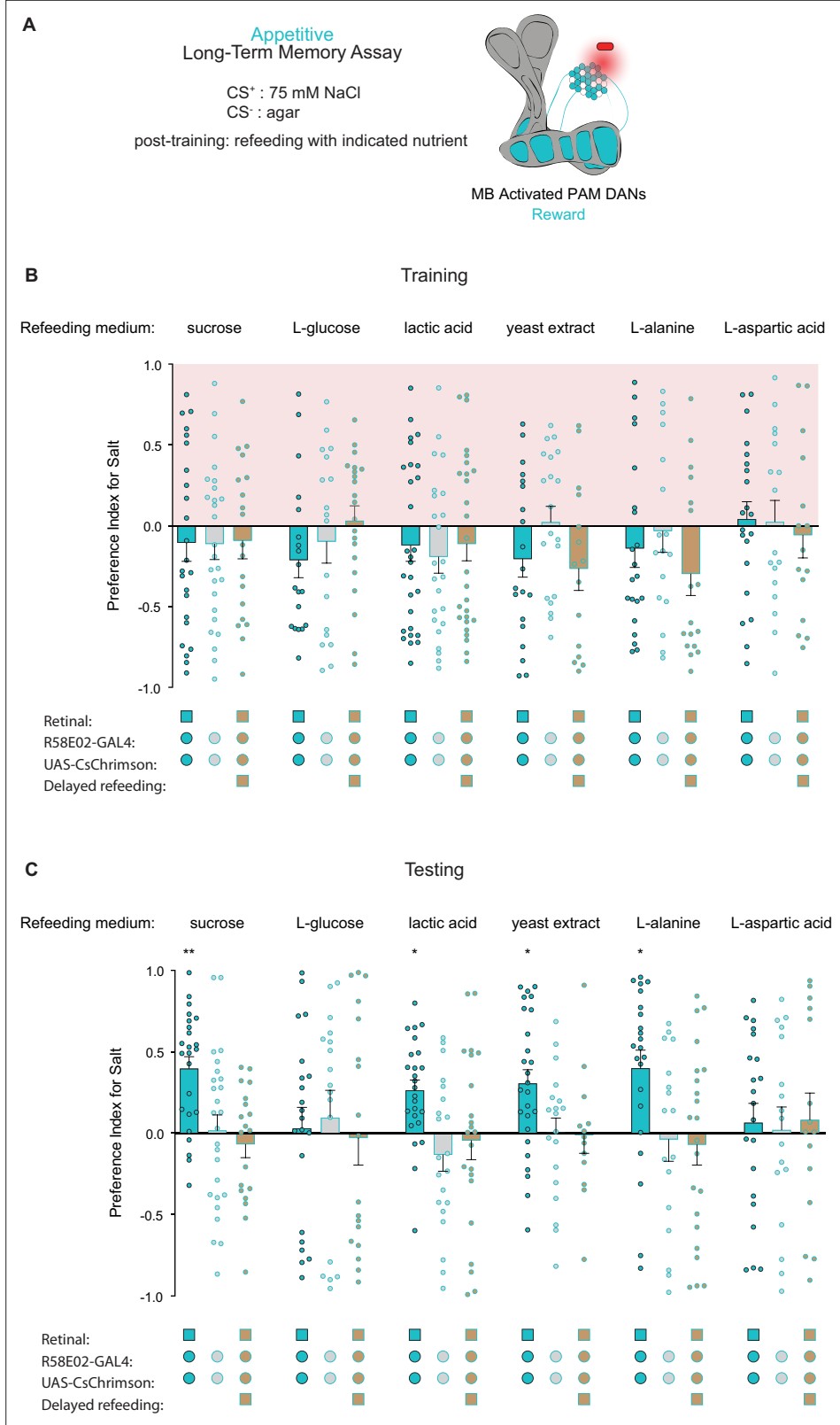

**Figure 5.** Caloric food sources are required for the formation of associative long-term taste memories. (**A**) Schematic of the conditions and mushroom body (MB) compartments innervated by broad protocerebral anterior medial (PAM) driver *R58E02-Gal4*. (**B, C**) Training (**B**) and testing (**C**) of appetitive long-term memories (LTM) measured following PAM neuron optogenetic activation paired with 75 mM NaCl (CS⁺) vs agar (CS⁻)

*Figure 5 continued on next page*

*Figure 5 continued*

(n=13–28). Flies were fed the indicated compounds for 1 hr immediately after training or after an 8 hr delay where indicated. All flies were starved for 24 hr prior to training. Preference indices are mean ± SEM, Kruskal-Wallis with Dunn's multiple comparison test: *p < 0.05, **p < 0.01.

The online version of this article includes the following source data for figure 5:

**Source data 1.** Raw data for *Figure 5*.

of PAM neurons during feeding modulates the strength of KC-MBON synaptic connections. Notably, activation of single PAM cell types produced different forms of memory in the STROBE. For example, stimulating PAM-α1 neurons during feeding drives appetitive taste LTM, while activation of PAM-β'1 was immediately rewarding.

The activation of bitter GRNs paired with sucrose led to the formation of STM, which agrees with previous research demonstrating that thermogenetic stimulation of bitter GRNs can negatively reinforce short-term taste learning (*Keene and Masek, 2012*). However, unlike sweet neuron activation, bitter neuron activation was not sufficient for the formation of LTM in our assay. One possible explanation is that the strong feeding inhibition evoked by bitter GRN activation leads to an insufficient number of CS-US pairings to induce LTM. Consistent with this idea, PPL1 activation, which induced LTM, is less aversive than bitter neuron activation during training, and therefore allows more associations.

A unique aspect of our long-term taste learning paradigm is that we uncoupled the US from a caloric food source. By doing this we were able to probe the energetic constraints gating LTM formation. It has long been reported that LTM formation in *Drosophila* requires the intake of caloric sugar. Here, we demonstrate that the caloric requirements of LTM formation can be fulfilled by food sources other than sucrose, including lactic acid and yeast extract. Moreover, it seems that at least one amino acid, L-alanine, is able to provide adequate energy, while others like L-aspartic acid cannot. We theorize that these foods may provide flies with readily accessible energy, as neurons are able to

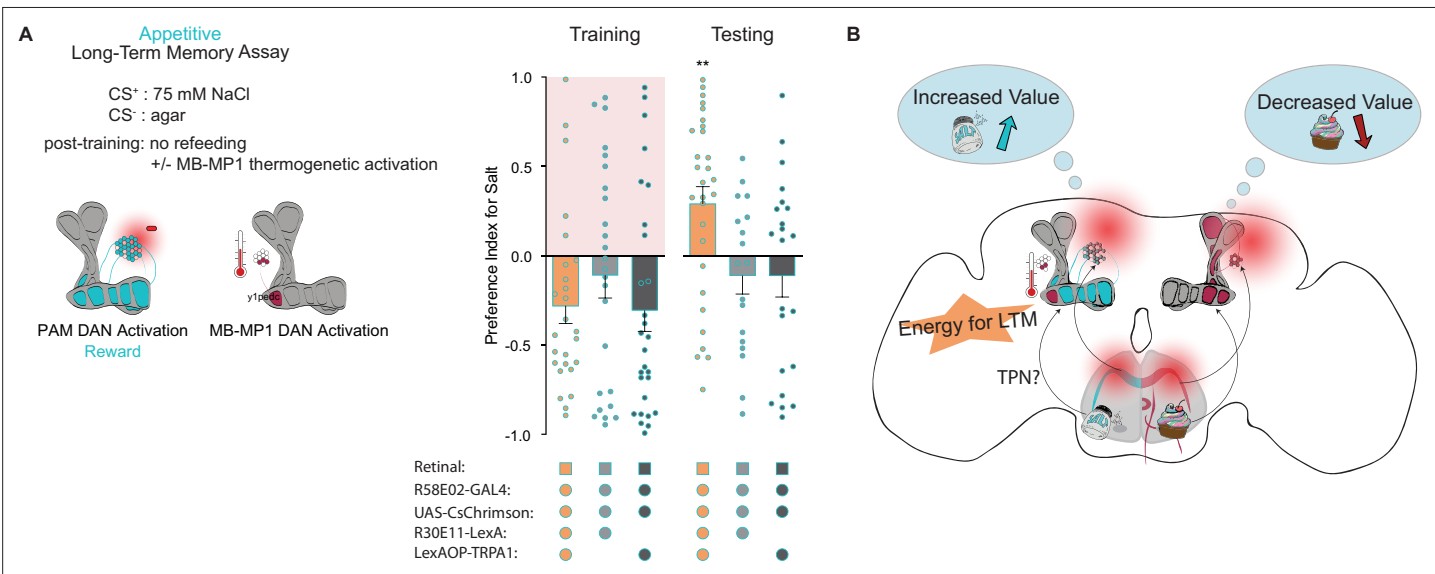

**Figure 6.** Mushroom body (MB)-MP1 neuron activation post training replaces energy signal required for the formation of long-term memories (LTM). (**A**) Appetitive LTM measured following protocerebral anterior medial (PAM) neuron optogenetic activation paired with 75 mM NaCl (CS⁺) vs agar (CS⁻) (n=18 = 29). Flies were transferred to 29°C for 1 hr immediately after training to thermogenetically activate MB-MP1 using *R30E11>TRPA1*. Controls are lacking one genetic element for MB-MP1 thermogenetic activation. Graphic of timeline followed for the LTM taste assay with thermogenetic activation of MB-MP1 neurons. All flies were starved for 24 hr prior to training. Preference indices are mean ± SEM, Kruskal-Wallis with Dunn's multiple comparison test: **p < 0.01.

The online version of this article includes the following source data for figure 6:

**Source data 1.** Raw data for *Figure 6*.

metabolize both lactic acid and L-alanine into pyruvate to fuel the production of ATP via oxidative phosphorylation (*de Tredern et al., 2021*).

Energy gating in the MB is thought to be regulated by the MB-MP1-DANs. MB-MP1 neuron oscillations activate increased mitochondrial energy flux within the KCs, which is both necessary and sufficient to support LTM (*Plaçais et al., 2017*). To demonstrate sufficiency in our assay, we activated MB-MP1 neurons with TRPA1 directly after fly training. This effectively substitutes for a caloric food source and allows LTM formation (*Figure 6C*). These results suggest that MB-MP1 neurons integrate energy signals during the formation of multiple types of LTM, and may be influenced by a variety of caloric foods.

Despite the advantages of replacing natural stimuli with optogenetic stimulation, there are also limitations. Most notably, optogenetic activation may not closely replicate temporal dynamics, intensity, or population features of natural stimulus encoding. Contact with food in the STROBE activates LED illumination with a relatively low latency of about 37±17 ms, but would not be expected to precisely mimic the onset of activation from natural taste stimuli (*Musso et al., 2019*). Moreover, bitter and acidic stimuli are known to evoke OFF responses that would not be replicated in the STROBE (*Devineni et al., 2021*; *Stanley et al., 2021*). Given the proposed importance of GRN temporal dynamics to higher-order neuronal plasticity, the critical role of timing between KC and DAN activation for MB plasticity, and the broad importance of timing to various synaptic plasticity mechanisms, it is easy to imagine that temporal differences between optogenetics and natural stimuli could differentially affect learning (*Cohn et al., 2015*; *Devineni et al., 2021*; *Handler et al., 2019*). Optogenetics and natural stimuli also undoubtedly activate different neuron populations. For example, *Gr64f-Gal4* labels most sweet sensory neurons, but the distribution of these neurons on different taste organs makes coincident activation of all these populations unlikely under natural conditions (*Fujii et al., 2015*). Conversely, direct DAN stimulation affects only a small subset of the neurons activated upon sugar taste detection and consumption, and likely therefore does not capture all of the effects that sugar has on appetitive conditioning (*Wang et al., 2004*). Nevertheless, the features of optogenetic activation are clearly sufficient to drive plasticity and learning.

Overall, our results suggest that lasting changes in the value of specific tastes can occur in response to temporal association with appetitive or aversive stimuli, raising the possibility that such plasticity plays an important role in animals' ongoing taste responses. It is interesting to speculate on what could serve as the US under more natural conditions. One obvious possibility is that pairing of different tastes (e.g. sugar and salt) in complex foods allows one taste to serve as the US and modifies future responses to the other. Intriguingly, tastes may also have the ability to self-reinforce over time, as shown for some odors (*Kato et al., 2022*). Another possibility is that natural association of tastes with non-taste reinforcers such as pain or mating could modify subsequent behavior. Future experiments using the STROBE paradigm could further probe the molecular and circuit mechanisms underlying taste memories and advance our understanding of how taste preferences may be shaped by experience over an animal's lifetime.

## Materials and methods
### Fly strains
Fly stocks were raised on a standard cornmeal diet at 25°C, 70% relative humidity. For neuronal activation, *20XUAS-IVS-CsChrimson.mVenus* (BDCS, stock number: 55135) was used. Dopaminergic PAM expression was targeted using previously described lines: *R58E02-GAL4* (*Musso et al., 2015*); *R58E02-LexA, R48B04-GAL4, R15A04-GAL4, R13F02-LexA,* and *R30E11-LexA* obtained from Bloomington (BDCS, stock numbers: 52740, 50347, 48671, 52460, 54209); and MB split-GAL4 lines *MB043B-GAL4, MB504B-GAL4, MB299B-GAL4, MB301B-GAL4* from Janelia Research Campus (*Aso et al., 2014*). GRN expression was driven using *Gr43a-GAL4, Gr64f-GAL4* (*Dahanukar et al., 2007*), *Gr66a-GAL4* (*Wang et al., 2004*), and *PPK23^glut^-GAL4, PPK23-GAL4, Gr66a-LexA::VP16, LexAop-Gal80* (*Jaeger et al., 2018*). *LexAop-tnt* was previously described (*Liu et al., 2016*). For temperature activation experiments, *LexAop-TrpA1* was used (*Liu et al., 2012*).

## STROBE experiments

Mated female *Drosophila* were collected 2–3 days post eclosion and transferred into vials containing 1 ml of standard cornmeal medium supplemented with 1 mM all-*trans*-retinal (Sigma #R2500) or an ethanol vehicle control. Flies were maintained on this diet for 2 days in a dark environment. 24 hr prior to experimentation, flies were starved at 25°C, 70% relative humidity, on 1% agar supplemented with 1 mM all-*trans*-retinal or ethanol vehicle control.

## STROBE training protocol

During the training phase for the STM experiments the STROBE was loaded with 4 µl of tastant (salt: Sigma #S7653 or sucrose: Sigma #S7903) on channel 1 and 4 µl 1% agar on channel 2. The red LED was triggered only when a fly interacted with the tastant in channel 1. The duration of the training period was 40 min. For the STM training protocol, flies were then transferred to clean empty vials for 10 min while the experimental apparatus was cleaned. The training and testing phases of LTM experiments were performed as described for the STM experiments with the following exception: after the 40 min training period flies were transferred individually into vials containing standard cornmeal diet or nutrient of interest in 1% agar (500 mM sucrose: Sigma #S7903, 500 mM L-glucose: Sigma #G5500, 250 mM lactic acid: Sigma #69785, 10% yeast extract: Sigma #Y1625, 250 mM L-alanine: Sigma #05129, 250 mM L-aspartic acid: Sigma #11230) and allowed to feed for 1 hr. They were then transferred into 1% agar starvation vials and kept at 18°C until the testing component of the experiment. For MB-MB1 activation experiments, after training flies were placed at 29°C, 70% relative humidity for 1 hr on 1% agar starvation vials. They were then transferred to 18°C and refed 8 hr later, outside of the memory consolidation. After 1 hr of feeding they were once again transferred into 1% agar starvation vials and kept at 18°C until the retrieval component of the experiment. The preference index for each individual fly was calculated as: (sips from channel 1 – sips from channel 2)/(sips from channel 1+sips from channel 2). All experiments were performed with a light intensity of 11.2 mW/cm$^2$ at 25°C, 70% relative humidity.

## STROBE testing protocol

During testing, 4 µl of the same tastant (salt: Sigma #S7653, sucrose: Sigma #S7903, MPG: Sigma #G1501) was reloaded into channel 1 and 4 µl of 1% agar on channel 2. The optogenetic component of the system was deactivated such that the red LED would no longer trigger if a fly interacted with the tastant. Flies were reloaded individually into the same arenas. The duration of the testing phase was 1 hr. The preference index for each individual fly was calculated as: (sips from channel 1 – sips from channel 2)/(sips from channel 1+sips from channel 2).

## Immunofluorescence microscopy

Brain staining protocols were performed as previously described (*Chu et al., 2014*). Briefly, brains were fixed for 1 hr in 4% paraformaldehyde and dissected in PBS + 0.1% Triton-X. After dissection brains were blocked in 5% NGS diluted with PBST for 1 hr. Brains were probed overnight at 4°C using the following primary antibody dilutions: rabbit anti-GFP (1:1000, Invitrogen #A11122, RRID: AB_221569) and mouse anti-brp (1:50, DSHB #nc82, RRID: AB_2392664). After a 1 hr wash period, secondary antibodies – goat anti-rabbit Alexa-488 (1:200, Invitrogen #A11008, RRID: AB_143165) and goat anti-mouse Alexa-568 (1:200, Invitrogen #A11030, RRID: AB_2534072) – were applied and incubated for 1 hr at room temperature to detect primary antibody binding. Slowfade gold was used as an antifade mounting medium.

Slides were imaged under a 25× water immersion objective using a Leica SP5 II Confocal microscope. All images were taken sequentially with a z-stack step size at 1 µm, a line average of 2, speed of 200 Hz, and a resolution of 1024×1024 pixels. ImageJ was used to compile slices into a maximum intensity projection (*Jaeger et al., 2018*).

## Statistical analysis

All statistical analyses were executed using GraphPad Prism 6 software. Sample size and statistical tests performed are provided in the Figure legends. Non-parametric tests were used because data did not always adhere to a normal distribution. For Dunn's multiple comparison tests, the experimental group was compared to all controls and the highest p-value reported over the experimental bar.

Replicates are biological replicates, using different individual flies from two or more crosses. Sample sizes were based on previous experiments in which effect size was determined. Data was excluded on the basis of STROBE technical malfunctions for individual flies and criteria for data exclusion are as follows: (i) if the light system was not working during training for individual arenas, (ii) if during training or testing a fly did not meet a standard minimum # of interactions for that genotype, (iii) if during training or testing the STROBE recorded an abnormally large # of interactions for that genotype, (iv) technical malfunctions due to high channel capacitance baseline activity, and (v) if a fly was dead in an arena.

## Code availability

All STROBE software is available for download from GitHub:
 FPGA code: https://github.com/rcwchan/STROBE-fpga ( *Chan, 2018a*).
 All other code: https://github.com/rcwchan/STROBE_software/ (*Chan, 2018b*).

## Acknowledgements

We thank Celia Lau for the original SEZ diagram models, and members of the Gordon lab for comments on the manuscript. This work was funded by Natural Sciences and Engineering Research Council (NSERC) grants RGPIN-2016-03857 and RGPAS 492846-16.

## Additional information

### Funding

| Funder | Grant reference number | Author |
|---|---|---|
| Natural Sciences and Engineering Research Council of Canada | RGPIN-2016-03857 | Michael D Gordon |
| Natural Sciences and Engineering Research Council of Canada | RGPAS 492846-16 | Michael D Gordon |

The funders had no role in study design, data collection and interpretation, or the decision to submit the work for publication.

### Author contributions

Meghan Jelen, Conceptualization, Formal analysis, Investigation, Visualization, Methodology, Writing – original draft, Writing – review and editing; Pierre-Yves Musso, Conceptualization, Methodology, Writing – review and editing; Pierre Junca, Methodology, Writing – review and editing; Michael D Gordon, Conceptualization, Supervision, Funding acquisition, Visualization, Writing – original draft, Writing – review and editing

### Author ORCIDs

Michael D Gordon http://orcid.org/0000-0002-5440-986X

### Decision letter and Author response

Decision letter https://doi.org/10.7554/eLife.81535.sa1
Author response https://doi.org/10.7554/eLife.81535.sa2

## Additional files

### Supplementary files

• MDAR checklist

### Data availability

All data generated or analyzed during this study are included in the manuscript; spreadsheets of raw numerical data are provided as source data files attached to each figure.

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
