## [Editor Report]

Through a new operant learning assay and fly genetics, this important work convincingly shows that taste memory formation requires the same circuit substrates and mechanisms as olfactory memory formation. While the exact mechanisms remain to be elucidated, the convincing data and approach represent a valuable foundation for the study of molecular and circuit mechanism underpinning taste memory formation and the role of brain energy therein. This study will be of particular interest to the large community of scientists studying the mechanisms and circuits of memory formation in the fly and possibly beyond.

---

## [Decision Letter]

**Decision letter after peer review:**

Thank you for submitting your article "Optogenetic induction of appetitive and aversive taste memories in *Drosophila*" for consideration by *eLife*. Your article has been reviewed by 3 peer reviewers, one of whom is a member of our Board of Reviewing Editors, and the evaluation has been overseen by Claude Desplan as the Senior Editor. The following individual involved in review of your submission has agreed to reveal their identity: Thomas Riemensperger (Reviewer #3).

Essential revisions:

All three reviewers expressed a high interest and, provided you can address the following issues, are supportive of publication of your manuscript in *eLife*. The reviewers were particularly impressed by the assay you have developed and its potential in resolving more dynamic aspects of perception and learning. In a revised version of the manuscript, please address especially the following points:

1. The reviewers recommend taking better advantage of your operant assay. Please show the data as original traces and not just box plots for the learning and recall phase. In addition, please explain why this is not 'simply' associative learning with one pairing as usually used in standard olfactory learning assays.

2. Explain the putative perceptual differences between optogenetic activation and real stimuli, for instance for sugar reward vs. exogenous activation of sweet taste neurons. Along the same lines, please show the data during learning as traces to help evaluate how actual stimulus perception during training impacts on learning.

3. Add reciprocal controls for at least one of the conditions.

4. We recommend making the manuscript more accessible to scientists not familiar with *Drosophila* memory paradigms by providing more explanation and interpretation of your data.

*Reviewer #1 (Recommendations for the authors):*

A few suggestions to improve the manuscript further:

Introduction:

(Line numbering would have been helpful…)

You write at the end of the first page, that DANs are activated by reward and punishment. They are also activated/modulated, for instance, by metabolic state and movement. Both of these aspects might be highly relevant in your operant assay.

End of p 3: 'while maintaining similar satiation states': this statement contrasts somewhat with some other statements you make regarding interaction with the food source and even the shown bar graphs during training. This needs to be taken into consideration and explained a little better.

Beginning of page 8 and Figures 5 and 6: please correct the references to and/or labeling of the panels. In addition, I did not find the data for the 7-hour delayed refeeding. Figure 6C does not exist.

In the schemes, the light dot is always shown only on one side of the brain. Is this correct that you are activating only unilaterally? I guess not and I found it somewhat confusing…

There were several typos or small mistakes that must be corrected.

*Reviewer #2 (Recommendations for the authors):*

Figures:

The authors describe the learning as a preference index, which is calculated from the number of interactions (sips) with the CSs. Does also the length of sip bouts change? Maybe it would be worth looking also at overall feeding time. This could be especially interesting as flies that have been trained with GR66a activation, still prefer the sugar CS over several minutes at the beginning of the test. Maybe these are just very short bouts. Is there an explanation for this phenotype? Are the flies maybe more food deprived, because they receive less sugar in the training?

It is unclear what L/R means in Figure 1C/E/G, as this is also not explained in the legend. I assume it is the left/right side of CSs. Could this be renamed as CS+/Cs-, or red light/ no red light?

The LLM/LTM can only be induced by feeding caloric food to the flies directly after training, this is a very important piece of information that does not show up in the figures. The authors should explain this fact a bit better in the text and maybe add this below the relevant graphs too (refeeding, yes/no), this would be especially helpful in Figure 6 where they do not receive food at all after training? For people not familiar with LTM this is hard to follow.

In Figure 3 some of the dots below the graphs do not seem to be in the right place. Please check 3A/B, they are different from 3C/D, but should be the same?

Discussion:

The paradigm that the authors present uses operant training, where the fly can decide how much they want to eat from each tastant presented. It would be great to have a paragraph in the discussion where the similarities and differences to previously studied classical conditioning are elucidated and results are compared. The authors could also cite work from the Brembs lab. In terms of reinforcement vs. US function of dopamine neurons, there was a recent biorxiv paper that could be cited if that is possible.: Pain is so close to pleasure: the same dopamine neurons can mediate approach and avoidance in *Drosophila* | bioRxiv Rohrsen et al., 2021.

*Reviewer #3 (Recommendations for the authors):*

The manuscript Optogenetic induction of appetitive and aversive taste memories in *Drosophila* by Jelen and colleagues is interesting, well written and the experiments elaborated, and the data provided by the authors justifies the title of the manuscript. The technique provided by the authors is very interesting and will attract a broad readership mainly of the *Drosophila* community as gustatory learning paradigms are yet tricky and not feasible for mass assays. The novel technique provided by Jelen and colleagues will pave the way to investigate gustatory learning now in more detail. In addition, the fact that the animals can freely move greatly improves the possibilities to investigate gustatory learning. Despite the high potential of the manuscript to reach out to a broad readership I would like to address some points of concern that the authors may want to address before publication.

In the first set of experiments the authors pair the artificial activation of bitter gustatory neurons with a low concentrated sugar solution as (CS+), whereas pure agar is presented as (CS-). To control for the learnability of a conditioned stimulus the authors have to ensure that the (CS-) and (CS+) are both equally learnable. Therefore, the authors should on the one hand provide first a naïve preference index between the two stimuli to demonstrate that both are equally perceived by the fly. On the other hand, the authors should equilibrate the conditioned stimuli in such a way that once agar is reinforced and once 25% sugar and display a learning index instead of the performance index. This of course should be done similarly for the NaCl conditioned stimulus.

It further appears that the (CS+) source is always situated on the left side of the experimental setup and the (CS-) always on the right side, if the indication of the interaction numbers are well understood. In order to avoid any place component that may interfere with the actual learning pathways the authors investigate the authors should also equilibrate the place of (CS+) presentation in relation to the (CS-).

According to many learning theories the learnability of a (CS+) is strongly increased when (CS+) and (US) are presented overlapping but with a time delay. Seen that the artificial activation of gustatory neurons together with the presentation of low-concentrated sugar may not only affect learning circuits but directly the perception of the (CS+) itself such a time delay would be even more important. Therefore, I suggest that the authors may introduce a time delay between the sipping and the opto-genetic activation of neurons in accordance with the published olfactory learning paradigms.

The authors need to explain the figures much more in detail. For example the given figure for figure 1C/D lets assume that the box blots in figure 1C reflect the data of the first ten minutes of the cumulative preference index. Indeed, this is apparently not the case, but the cumulative preference index is depicted over the entire 60 min but 10 min after the training. The figure description is misleading and would need some amendments. Further the authors should explain more in detail what the interaction values are and what they reflect. The information given by the authors is cryptic and does not allow a straight-forward understanding of the figure.

As stated above the cumulative preference index indicates a strong delay between the two groups in their memory retrieval (Figure 1D). The authors do not really discuss this effect in detail that per se is very interesting as it is very different in its dynamics compared with the other learning experiments provided by the authors.

For the statistical analysis the authors use an ANOVA with a Dunnet's post hoc correction throughout. In this regard it is unclear which of the data groups serve as a reference for the test. Normally a Dunnet's correction is used for multiple test groups that are compared to one single control group e.g., Placebo against different medical treatments. Here the Placebo group would serve as reference. In the case of the data provided by the authors, the situation is drastically different, as we have one test group and three control groups. As such the Dunnet's correction may not be the most adequate way for a multiple comparison of data and the authors may want to think about employing a more standard correction such as Bonferroni or Tukey.

The authors use two terms when referring to forms of memory exceeding short-term memory, long-term memory (LTM) and long lasting memory (LLM). However, they miss to explain when and why they employ the two different terms.

Further, the authors should help the reader and indicate more rigorously the compartments that are innervated by the individual lines. Descriptions like R48B04>CsChrimson or the "activation of R15A04-Gal4 neuron" are difficult to follow for readers that not directly related to the field.

Lastly, I would like to encourage the authors to employ their intuitive technique to expand the field of gustatory learning instead of asking questions that were already answered for olfactory conditioning now for gustatory conditioning. Of course, it is interesting to see the parallels between gustation and olfaction but the cellular mechanisms and energy availability would rather be a surprise if they would differ in their mode of action between the two forms of learning. However, the technique described by Jelen and colleagues would allow much more detailed circuit-oriented and temporal analysis of gustatory learning.

---

## [Author Response]

Essential revisions:All three reviewers expressed a high interest and, provided you can address the following issues, are supportive of publication of your manuscript in eLife. The reviewers were particularly impressed by the assay you have developed and its potential in resolving more dynamic aspects of perception and learning. In a revised version of the manuscript, please address especially the following points:1. The reviewers recommend taking better advantage of your operant assay. Please show the data as original traces and not just box plots for the learning and recall phase. In addition, please explain why this is not 'simply' associative learning with one pairing as usually used in standard olfactory learning assays.

All graphs now include raw data points in addition to the summary statistics. In the text we now also note and comment on the individual variability seen in the data.

We have added a section to the introduction in which we note the operant nature of our paradigm and how it differs from classical conditioning (lines 125-130). We have also added a paragraph to the Discussion where we comment on our results in the context of the differences between operant and classical conditioning, as well as past studies exploring relationships between these two types of learning in flies (lines 340-352).

2. Explain the putative perceptual differences between optogenetic activation and real stimuli, for instance for sugar reward vs. exogenous activation of sweet taste neurons. Along the same lines, please show the data during learning as traces to help evaluate how actual stimulus perception during training impacts on learning.

We have added a paragraph to the Discussion in which we describe some limitations of our approach, with particular focus on the ways that optogenetic stimulation may not replicate all the features of natural stimuli (lines 409-427).

We have also added time curves showing the evolution of preference for the two stimuli during both training and testing of all the experiments in Figures 1 and 2. Since the rest of the paper explores manipulations that derive from the basics laid out in Figures 1 and 2, we decided that it is sufficient to show the temporal dynamics for only those earlier experiments.

3. Add reciprocal controls for at least one of the conditions.

We have now expanded Figure 2—figure supplement 1 and its associated text in order to present and describe several controls to demonstrate the specificity of learning the CS+ in our paradigm. In order to avoid redundancy, we will discuss these additions in more detail response to the more specific reviewer comments below.

4. We recommend making the manuscript more accessible to scientists not familiar with *Drosophila* memory paradigms by providing more explanation and interpretation of your data.

We appreciate this suggestion and have extensively rewritten the manuscript to add clarity and accessibility to both the background material and also our results. We have also reworked the figures and figure legends to increase clarity.

Reviewer #1 (Recommendations for the authors):A few suggestions to improve the manuscript further:Introduction:(Line numbering would have been helpful…)You write at the end of the first page, that DANs are activated by reward and punishment. They are also activated/modulated, for instance, by metabolic state and movement. Both of these aspects might be highly relevant in your operant assay.

We appreciate you mentioning this and agree that this is relevant for operant learning. However, in rewriting this section for clarity, we have now explicitly introduced the DANs in the context of olfactory classical conditioning where their activity is usually evoked by sugar or shock. However, we have added a passage near the end of the Discussion (lines 431-437) where we briefly discuss what could function to activate DANs during natural taste learning.

End of p 3: 'while maintaining similar satiation states': this statement contrasts somewhat with some other statements you make regarding interaction with the food source and even the shown bar graphs during training. This needs to be taken into consideration and explained a little better.

This is an excellent point. Our intention was to point out that changes in satiety could influence behavior in a way that appears to be learning, whereas in our experiments we point out that any differences in feeding during training are actually in the opposite direction to what could explain the change in behavior (e.g. flies eating more salt during training and then still showing increased salt feeding during testing). Nonetheless, during revision of the text for clarity we have removed the statement altogether.

Beginning of page 8 and Figures 5 and 6: please correct the references to and/or labeling of the panels. In addition, I did not find the data for the 7-hour delayed refeeding. Figure 6C does not exist.

The delayed refeeding is indicated for each nutrient in the figure (gold bars). We have corrected the other errors.

In the schemes, the light dot is always shown only on one side of the brain. Is this correct that you are activating only unilaterally? I guess not and I found it somewhat confusing…

We have tried, where possible, to expand the red light to suggest global illumination (which is the case).

There were several typos or small mistakes that must be corrected.

We have extensively rewritten and edited the manuscript, so we hope to have caught everything.

Reviewer #2 (Recommendations for the authors):Figures:The authors describe the learning as a preference index, which is calculated from the number of interactions (sips) with the CSs. Does also the length of sip bouts change? Maybe it would be worth looking also at overall feeding time. This could be especially interesting as flies that have been trained with GR66a activation, still prefer the sugar CS over several minutes at the beginning of the test. Maybe these are just very short bouts. Is there an explanation for this phenotype? Are the flies maybe more food deprived, because they receive less sugar in the training?

These are interesting questions but unfortunately the code we use to run the STROBE does not collect data on sip length, bout duration, or any of the other more detailed metrics that are possible with the original FlyPad code.

It is certainly likely that flies trained with Gr66a activation are more food deprived than controls. However, we also caution (as discussed above) against interpreting preference indices early in the assay. At these early time points the flies have made few sips and the overall preference measurement is likely to not be very reliable.

It is unclear what L/R means in Figure 1C/E/G, as this is also not explained in the legend. I assume it is the left/right side of CSs. Could this be renamed as CS+/Cs-, or red light/ no red light?

We have changed these to “agar” and “salt” or “suc” to indicate the food option in these figures.

The LLM/LTM can only be induced by feeding caloric food to the flies directly after training, this is a very important piece of information that does not show up in the figures. The authors should explain this fact a bit better in the text and maybe add this below the relevant graphs too (refeeding, yes/no), this would be especially helpful in Figure 6 where they do not receive food at all after training? For people not familiar with LTM this is hard to follow.

We now have indicated the refeeding status for each LTM figure in the schematic associated with the figure.

In Figure 3 some of the dots below the graphs do not seem to be in the right place. Please check 3A/B, they are different from 3C/D, but should be the same?

Thank you for this very important catch. We have corrected the error.

Discussion:The paradigm that the authors present uses operant training, where the fly can decide how much they want to eat from each tastant presented. It would be great to have a paragraph in the discussion where the similarities and differences to previously studied classical conditioning are elucidated and results are compared. The authors could also cite work from the Brembs lab. In terms of reinforcement vs. US function of dopamine neurons, there was a recent biorxiv paper that could be cited if that is possible.: Pain is so close to pleasure: the same dopamine neurons can mediate approach and avoidance in *Drosophila* | bioRxiv Rohrsen et al., 2021.

Thank you for these very insightful suggestions. We have added a paragraph where we discuss the interpretation of our data in light of the possible contributions from operant and classical conditioning and what is known about these two forms of learning from important earlier work from Brembs. We have also added the suggested reference in the results when noting the differential effects of different PAM subpopulations.

Reviewer #3 (Recommendations for the authors):The manuscript Optogenetic induction of appetitive and aversive taste memories in *Drosophila* by Jelen and colleagues is interesting, well written and the experiments elaborated, and the data provided by the authors justifies the title of the manuscript. The technique provided by the authors is very interesting and will attract a broad readership mainly of the *Drosophila* community as gustatory learning paradigms are yet tricky and not feasible for mass assays. The novel technique provided by Jelen and colleagues will pave the way to investigate gustatory learning now in more detail. In addition, the fact that the animals can freely move greatly improves the possibilities to investigate gustatory learning. Despite the high potential of the manuscript to reach out to a broad readership I would like to address some points of concern that the authors may want to address before publication.In the first set of experiments the authors pair the artificial activation of bitter gustatory neurons with a low concentrated sugar solution as (CS+), whereas pure agar is presented as (CS-). To control for the learnability of a conditioned stimulus the authors have to ensure that the (CS-) and (CS+) are both equally learnable. Therefore, the authors should on the one hand provide first a naïve preference index between the two stimuli to demonstrate that both are equally perceived by the fly. On the other hand, the authors should equilibrate the conditioned stimuli in such a way that once agar is reinforced and once 25% sugar and display a learning index instead of the performance index. This of course should be done similarly for the NaCl conditioned stimulus.

We have now expanded Figure 2—figure supplement 1 to include a number of controls to address the issues raised. We chose to focus on using NaCl and MPG as the conditioned stimuli, with PAM activation serving to drive appetitive memory formation. We show first that the naïve preference for NaCl vs MPG is roughly equal (panel C). Next, we show that, like NaCl, flies are able to learn to prefer MPG over agar when MPG is trained as the CS+ (panel D). We also show that an appetitive memory for NaCl can be formed when NaCl is the CS+ and MPG is the CS-. Finally, we show that the memory formed when NaCl is the CS+ is not generalized to MPG, as training with NaCl as the CS+ does not result in an elevated preference for MPG vs agar (panel E). Although we acknowledge that these are not the exact experiments requested, we were very limited in the number of additional experiments we were able to perform, and feel that these demonstrate the same qualities that the reviewer is looking for in their comment.

It further appears that the (CS+) source is always situated on the left side of the experimental setup and the (CS-) always on the right side, if the indication of the interaction numbers are well understood. In order to avoid any place component that may interfere with the actual learning pathways the authors investigate the authors should also equilibrate the place of (CS+) presentation in relation to the (CS-).

In panels A and B of Figure 2—figure supplement 1, we now demonstrate that flies trained with NaCl as the CS+ do not exhibit a positional preference when agar is presented on both sides during testing. This is true whether the training is done with PAM activation (panel A) or sweet neuron activation (panel B). We performed both these experiments since sweet neuron activation drives a positive preference during training, while PAM activation does not. Along with the other experiments described above, these results demonstrate the specificity of training to the CS+ in the STROBE paradigm.

According to many learning theories the learnability of a (CS+) is strongly increased when (CS+) and (US) are presented overlapping but with a time delay. Seen that the artificial activation of gustatory neurons together with the presentation of low-concentrated sugar may not only affect learning circuits but directly the perception of the (CS+) itself such a time delay would be even more important. Therefore, I suggest that the authors may introduce a time delay between the sipping and the opto-genetic activation of neurons in accordance with the published olfactory learning paradigms.

This is a great idea and would be an excellent way to leverage the STROBE to manipulate different temporal dynamics during training. However, we do not currently have the expertise in the lab to modify the sip detection and light triggering algorithm to introduce a systematic delay. Nonetheless, we should point out that, while the STROBE LED activates with low latency, there is still a delay between sips and LED illumination. We previously measured this delay to be about 37 ms.

The authors need to explain the figures much more in detail. For example the given figure for figure 1C/D lets assume that the box blots in figure 1C reflect the data of the first ten minutes of the cumulative preference index. Indeed, this is apparently not the case, but the cumulative preference index is depicted over the entire 60 min but 10 min after the training. The figure description is misleading and would need some amendments. Further the authors should explain more in detail what the interaction values are and what they reflect. The information given by the authors is cryptic and does not allow a straight-forward understanding of the figure.

We have now modified all the figures and figure legends with the goal of increasing clarity. We paid particular attention to the schematics and the presentation of timing. We have removed the timeline schematic for each figure, as they were redundant and, upon reflection, confusing for several reasons. We now rely on Figure 1A to clearly demonstrate the paradigm and the timing for both STM and LTM assays. The schematics associated with individual panels now illustrate only the neuron population being optogenetically activated and the identity of the CS+ and CS-, as these are the parameters that change from experiment to experiment.

As stated above the cumulative preference index indicates a strong delay between the two groups in their memory retrieval (Figure 1D). The authors do not really discuss this effect in detail that per se is very interesting as it is very different in its dynamics compared with the other learning experiments provided by the authors.

As described in response to another reviewer’s comment above, we believe that the calculated preference indices are relatively unreliable in the early portion of testing, due to the small number of interactions during that period and the stochastic effects of flies encountering one choice vs the other. Thus, we caution against reading too much into changes in preference during these early times. Nonetheless, it is possible that the control group develops a stronger preference for sugar over the course of the assay as some hunger builds up after their feeding during training, and this combines with higher reliability as sips build up to drive a higher preference. We now mention our general interpretation of the testing dynamics in the text.

For the statistical analysis the authors use an ANOVA with a Dunnet's post hoc correction throughout. In this regard it is unclear which of the data groups serve as a reference for the test. Normally a Dunnet's correction is used for multiple test groups that are compared to one single control group e.g., Placebo against different medical treatments. Here the Placebo group would serve as reference. In the case of the data provided by the authors, the situation is drastically different, as we have one test group and three control groups. As such the Dunnet's correction may not be the most adequate way for a multiple comparison of data and the authors may want to think about employing a more standard correction such as Bonferroni or Tukey.

From a statistical standpoint, we struggle to see the distinction between comparing one control to several test groups and comparing one test group to several controls. The similarity between these situations is exactly why we chose the Dunnet’s test, and then report the lowest level of significance (highest p-value) across the comparisons with the different controls. Nonetheless, we tested several experiments with using a Bonferroni test instead of the Dunnet’s test and there was no difference in the categorical result (i.e. the number of stars). In the end we elected to continue with the Dunnet’s test.

The authors use two terms when referring to forms of memory exceeding short-term memory, long-term memory (LTM) and long lasting memory (LLM). However, they miss to explain when and why they employ the two different terms.

After further consideration, we have decided to simply use LTM throughout. Originally we used LLM in the portion of the manuscript preceding the demonstration that the memory is protein synthesis dependent. However, we agree this is unnecessarily confusing and have elected to simplify by only using LTM.

Further, the authors should help the reader and indicate more rigorously the compartments that are innervated by the individual lines. Descriptions like R48B04>CsChrimson or the "activation of R15A04-Gal4 neuron" are difficult to follow for readers that not directly related to the field.

We have now added this information to the text as we describe each experiment.

Lastly, I would like to encourage the authors to employ their intuitive technique to expand the field of gustatory learning instead of asking questions that were already answered for olfactory conditioning now for gustatory conditioning. Of course, it is interesting to see the parallels between gustation and olfaction but the cellular mechanisms and energy availability would rather be a surprise if they would differ in their mode of action between the two forms of learning. However, the technique described by Jelen and colleagues would allow much more detailed circuit-oriented and temporal analysis of gustatory learning.

We appreciate this suggestion and the sentiment behind it. As we performed all of the experiments presented in the manuscript, we were always pulled between the desire to demonstrate the robustness and utility of the assay versus the desire to tread new ground and explore major differences in learning mechanisms. For us, each result that confirmed a similarity with olfactory learning was additional evidence that the assay was really measuring what we thought it was measuring, and this contributed to the appeal of covering all the similarities with olfactory learning. As with many projects, circumstances (in particular the pandemic in this case) also contributed to how much we could achieve. Nonetheless, we look forward to probing more taste learning mechanisms in the future.